# Neglected Spleen Transcriptional Profile Reveals Inflammatory Disorder Conferred by Rabbit Hemorrhagic Disease Virus 2 Infection

**DOI:** 10.3390/v16040495

**Published:** 2024-03-23

**Authors:** Jifeng Yu, Yan Li, Lu Xiao, Jing Xie, Zhiqiang Guo, Yonggang Ye, Yi Lin, Ye Cao, Xuejing Wu, Congjian Mao, Xingyu Li, Meng Pan, Jianqiang Ye, Long Zhou, Jian Huang, Junyan Yang, Yong Wei, Xianhui Zhang, Bin Zhang, Runmin Kang

**Affiliations:** 1Sichuan Provincial Key Laboratory of Animal Breeding and Genetics, Sichuan Animal Science Academy, Chengdu 610066, China; yujifeng1919@163.com (J.Y.); yaanxiaolu@163.com (L.X.); xiemm75@163.com (J.X.); ygzhiq@126.com (Z.G.); yeyg0202@sina.com (Y.Y.); linyi-sc@163.com (Y.L.); cy3831@163.com (Y.C.); xuejingwuchn@163.com (X.W.); maocongjian913@163.com (C.M.); yuxing0219@163.com (X.L.); panmeng202402@163.com (M.P.); yjq3240247@126.com (J.Y.); yyyjy959299@163.com (J.Y.); veishangyan@163.com (Y.W.); zhangxianhui202402@163.com (X.Z.); 2College of Animal Science and Veterinary Medicine, Southwest Minzu University, Chengdu 610041, China; lyvet2004@163.com (Y.L.); zhoulongscu@163.com (L.Z.); huangjian.1122@163.com (J.H.); binovy@sina.com (B.Z.)

**Keywords:** rabbit hemorrhagic disease virus 2 (RHDV2), spleen, pathogenicity, transcriptomic analysis

## Abstract

Rabbit hemorrhagic disease (RHD) is an acute fatal disease caused by the rabbit hemorrhagic disease virus (RHDV). Since the first outbreaks of type 2 RHDV (RHDV2) in April 2020 in China, the persistence of this virus in the rabbit population has caused substantial economic losses in rabbit husbandry. Previous failures in preventing RHDV2 prompted us to further investigate the immune mechanisms underlying the virus’s pathogenicity, particularly concerning the spleen, a vital component of the mononuclear phagocyte system (MPS). For this, a previous RHDV2 isolate, CHN/SC2020, was utilized to challenge naive adult rabbits. Then, the splenic transcriptome was determined by RNA-Seq. This study showed that the infected adult rabbits had 3148 differentially expressed genes (DEGs), which were associated with disease, signal transduction, cellular processes, and cytokine signaling categories. Of these, 100 upregulated DEGs were involved in inflammatory factors such as IL1α, IL-6, and IL-8. Kyoto Encyclopedia of Genes and Genomes (KEGG) analysis showed that these DEGs were significantly enriched in the cytokine–cytokine receptor interaction signaling pathway, which may play a vital role in CHN/SC2020 infection. At the same time, proinflammatory cytokines and chemokines were significantly increased in the spleen at the late stages of infection. These findings suggested that RHDV2 (CHN/SC2020) might induce dysregulation of the cytokine network and compromise splenic immunity against viral infection, which expanded our understanding of RHDV2 pathogenicity.

## 1. Introduction

Rabbit hemorrhagic disease (RHD) is a highly contagious, acute, fatal infectious disease that affects wild and domestic rabbits [1]. The causative agent, the RHD virus (RHDV), was first identified in 1984 in China [2] and rapidly spread worldwide within a few years. According to the taxonomy of viruses, RHDV, belonging to the genus lagovirus within the family Caliciviridae [3], is a non-enveloped virus that contains a single-stranded RNA genome of approximately 7.4 kilobases (kb) [4]. The genome of RHDV encodes two slightly overlapping open reading frames (ORFs), ORF1 and ORF2. ORF1 encodes a large polyprotein, which is cleaved by a virus-encoded protease to generate seven non-structural proteins (NSPs) and the major capsid protein (VP60), while ORF2 produces a minor structural protein (VP10) [5,6].

Based on the global RHDV classification system and VP60 sequence, pathogenic RHDV strains were divided into three distinct groups: classic RHDV (G1–G5), the antigenic variant RHDVa (G6), and the new variant RHDV2 (RHDVb/GI.2) [7,8,9,10,11,12]. The re-emerging rabbit hemorrhagic disease virus type 2 (RHDV2) caused an unexpected RHD outbreak in April 2020 in Sichuan Province, Southwest China [13]. Then, the outbreak rapidly spread among rabbit populations across broad geographic areas [14]. This outbreak was a suspected foreign invasion of RHDV2 due to its high nucleotide homology with Dutch isolates in 2016, which implicated its robust viral genetic fitness to damage the host immune defense in rabbit colonies [15,16].

Several lines of evidence suggested that the compromised host immunity of naive rabbits promoted their susceptibility to RHDV2 infection, resulting in varying clinical outcomes [17,18]. The molecular, histological, and immunological profiles of infected rabbits suggested the role of host immunity in resistance against RHDV2 invasion, especially by the heterogeneous activation of the mononuclear phagocyte system (MPS) [19,20]. The spleen, containing abundant lymphocytes and phagocytes, was a crucial component of the MPS and mediated the humoral and cellular immunity of rabbits [19,20]. The transcriptional investigation of the spleen following RHDV2 infection had never been conducted, and its role in virus–host interaction remains unknown.

A few early studies focused on the liver tropism of RHDV2, virus burdens, and the serology dynamic during the infection and vaccine cross-protection [21,22]. Molecular pathways for virus entry, replication, and perturbation of inflammatory activities were highlighted by multi-omics analysis, which indicated the interactive networks of antigen recognition and presentation by MPS in early and late RHDV2 infection [18,23,24]. It was speculated that lagoviruses might trade with immune cells to escape innate and adaptive immunity through specific, unexplained mechanisms.

Our previous study investigated an RHDV2 isolate, called RHDV2-CHN/SC2020, for its genome characteristics and pathogenicity [16,25], which frequently injures the liver and spleen, especially in young rabbits. RHDV2 significantly compromised the innate immunity of young rabbits compared to RHDV1 in terms of the early liver transcriptional spectrum [18]. Furthermore, there was no significant difference in tissue viral load or pathological damage between young and adult rabbits infected with RHDV2. We speculated that the spleen, which harbored a substantial population of lymphocytes and phagocytes, was part of a robust MPS arsenal and participated in the trade of RHDV2 with extrahepatic host immunity (humoral and cellular immunity), thereby promoting viral immune evasion. Thus, the spleen transcriptional profile might help us expand our knowledge of RHDV2 pathogenicity in rabbits. Therefore, this study aimed to explore the spleen’s activated immune network responding to the RHDV2 invasion and determine the pivotal molecules contributing to RHDV2 infection and persistence.

## 2. Materials and Methods

### 2.1. Viruses and Animals

The RHDV2-CHN/SC2020 strain (GenBank no. MT434995) was isolated from domestic rabbits that succumbed to RHD in April 2020 in Sichuan, China. Sixteen New Zealand white rabbits were supplied by Chengdu Dossy Experimental Animals CO., LTD (Chengdu, China). The rabbits were fed under standard housing conditions with unrestricted access to food and water. The rabbits were handled according to the Standards for Laboratory Animals (GB14925-2010) and Guideline on the Humane Treatment of Laboratory Animals (MOST 2006a) of China. All rabbits were raised in the Experimental Animal Center of the Sichuan Animal Science Academy (Chengdu, China).

### 2.2. Infection and Collection of Biological Samples

Ten clinically healthy New Zealand white rabbits (10-week-old) were confirmed to be negative for antibodies to RHDV1 and RHDV2 (HI < 2^3^). In the infection group, five rabbits were infected subcutaneously with 2 × 10^8^ capsid gene copies of CHN/SC2020 on day 0. Five control rabbits were sham inoculated with 1 mL of phosphate-buffered saline (PBS). The clinical course of the infection was recorded daily. Liver, spleen, lung, kidney, and heart samples were collected from the rabbits at the moribund stage. One part of each sample was fixed in 10% neutral-buffered formalin. The rest of the samples were directly stored at −80 °C until processing.

### 2.3. Histopathological Observation

The tissue samples fixed in 10% neutral-buffered formalin for 24 h were paraffin-embedded, sectioned at 4 μm, and then stained with hematoxylin and eosin. The histopathology of the sample section was observed under a light microscope (Motic, Germany).

### 2.4. Virus Quantification

The tissue samples from each rabbit were homogenized with a tissue master. Total RNA was then extracted using Trizol RNAiso plus (TaKaRa, Dalian, China) according to the manufacturer’s protocol. Viral load was quantified in terms of the “capsid gene copy number” by using the quantitative reverse transcription polymerase chain reaction (qRT-PCR) method, which specifically detected VP60 of RHDV2. Information on all primers is provided in Appendix A.

### 2.5. cDNA Library Construction and Sequencing

Six New Zealand white rabbits (10-week-old) were divided into an infected group and a control group, and three rabbits were infected with CHN/SC2020 according to the above operations. Then, the spleens were collected at 16 h post infection (hpi). The spleens of the control group were also collected at the same time. About 100 mg of each sample was homogenized with a tissue master. Total RNA was isolated using TRIzol reagent (Invitrogen, Carlsbad, CA, USA). The total RNA concentration was determined by measuring the absorbance at 260 nm. The quality of the RNA samples was verified using RNA Nano 6000 Assay Kit of the Agilent 2100 Bioanalyzer system (Agilent Technologies, Santa Clara, CA, USA) prior to further processing.

Six RNA-seq libraries were prepared from 200 ng of total RNA using a NEBNext^®^ Ultra™ II RNA Library Prep Kit (NEB, Boston, MA, USA) according to the manufacturer’s instructions. Briefly, mRNA was purified from total RNA using poly-T oligo-attached magnetic beads. Fragmentation was carried out using divalent cations at elevated temperatures in the First-Strand Synthesis Reaction Buffer (5×). First-strand cDNA was synthesized using a random hexamer primer and M-MuLV Reverse Transcriptase (RNase H-). Second-strand cDNA synthesis was subsequently performed using DNA Polymerase I and RNase H. Remaining overhangs were converted into blunt ends via exonuclease/polymerase activities. After adenylation of the 3′ ends of DNA fragments, adaptors with a hairpin loop structure were ligated to prepare for hybridization. In order to select cDNA fragments that were preferentially 370~420 bp in length, the library fragments were purified with the AMPure XP system (Beckman Coulter, Beverly, MA, USA). Then, PCR was performed with Phusion High-Fidelity DNA polymerase, Universal PCR primers, and Index (X) Primer. Finally, PCR products were purified (AMPure XP system, Beckman Coulter, Brea, CA, USA), and library quality was assessed on the Agilent Bioanalyzer 2100 system (Agilent Technologies, Santa Clara, CA, USA). The clustering of the index-coded samples was performed on a cBot Cluster Generation System using TruSeq PE Cluster Kit v3-cBot-HS (Illumina) according to the manufacturer’s instructions. After cluster generation, the library preparations were sequenced on an Illumina Novaseq 6000 platform, and 150-bp paired-end reads were generated.

### 2.6. Pre-Processing of Sequencing Reads and Gene Expression and Differential Gene Analyses

Raw data (raw reads) in fastq format were first processed through fastp software. In this step, clean data (clean reads) were obtained by removing reads containing adapter, reads containing ploy-N, and low-quality reads from raw data. The clean reads were aligned with the oryctolagus cuniculus genome (https://www.ncbi.nlm.nih.gov/datasets/genome/GCF_009806435.1/ (accessed on 4 February 2021)) using Hisat2 (version 2.0.5). The oryctolagus cuniculus genome gene annotation (NCBI release) was also downloaded from the National Center for Biotechnology Information’s website. FeatureCounts v1.5.0-p3 was used to count the read numbers mapped to each gene. And then the expected number of fragments per kilobase of transcript per million fragments sequenced (FPKM) of each gene was calculated based on the length of the gene and read count mapped to this gene. Principal component analysis (PCA) can be used to reveal gene expression differences in biological samples based on the R language ggplot2 package, and this approach was used to analyze the sample data from the spleen. For the biological duplicate samples, Deseq2 was used to calculate the log_2_ fold-change (Log_2_FC) and probability for each gene in every comparison using strict criteria (Log_2_FC > 1 or Log_2_FC < −1, *p.adj* < 0.05).

### 2.7. Gene Ontology (GO) and Pathway Enrichment Analysis

GO enrichment analysis of differentially expressed genes was implemented by the clusterProfiler R package, in which gene length bias was corrected. GO terms with a corrected P value less than 0.05 were considered significantly enriched by differentially expressed genes. All of the GO annotation information was obtained from the Nr database, and we used GO: TermFinder (https://go.princeton.edu/cgi-bin/GOTermFinder (accessed on 21 September 2023)) to obtain information about the gene classes. Statistical analyses relating to the hypergeometric test and the FDR method were conducted using the R package, and all the GO analyses used a custom-made Perl script. Pathway enrichment analysis was performed using KAAS (KEGG Automatic Annotation Server, http://www.genome.jp/tools/kaas/ (accessed on 21 September 2023)) to functionally annotate the genes using BLAST comparisons against the manually curated KEGG database. We used clusterProfiler R package to test the statistical enrichment of differential expression genes in KEGG pathways.

### 2.8. Quantitative Real-Time Polymerase Chain Reaction (qRT-PCR) Verification of the Illumina NovaSeq 6000 Sequencing Data

Differentially expressed genes (DEGs) were examined by established qRT-PCR methods in present study or previous studies [26,27,28,29] to confirm the accuracy of the sequencing data. The primer sequences for the nine selected DEGs are listed in Appendix A. The total RNA extracted from the spleen of infected and non-infected rabbits using TRIzol reagent (TaKaRa, Dalian, China) was reverse transcribed using the PrimeScript™ RT reagent Kit (TaKaRa, Dalian, China). For the qRT-PCR analysis, the TB Green^®^ Premix Ex Taq™ II (Tli RNaseH Plus) Kit (TaKaRa, Dalian, China) and the QuantStudio 3 real-time PCR system (ThermoFisher, Singapore) were used according to each manufacturer’s instructions. The relative expression level of each gene was calculated using the 2^−ΔΔCt^ method.

### 2.9. Statistical Analysis

The differences among all categorized variables were analyzed using Student’s *t*-test. A *p*-value of 0.05 indicates statistical significance.

## 3. Results

### 3.1. Clinical Signs, Case Fatality Rates, and Histopathological Changes in Experimentally Infected Rabbits

After being challenged with the CHN/SC2020 strain, five rabbits showed RHD signs, including anorexia and lethargy, and two rabbits bled from the nose (Appendix A). All infected rabbits died within 24 hpi. Lesions at necropsy were typical of RHD, including hemorrhagic liver, lungs, and trachea (Appendix A). The spleens of dead rabbits were significantly enlarged (Appendix A). The survival time and mortality rate were displayed in Table 1. The viral load of the liver had an average value of 1.98 × 10^8^ per 0.1 g of tissues. No RHD signs or mortality were observed in the uninfected controls. At the end of the trial, no macroscopic lesions were observed at necropsy, and the tissue samples collected from the control rabbits were free of RHDV2 RNA (Appendix A).

Tissues were prepared for histopathological observation to evaluate the histocytic damage caused by CHN/SC2020 infection in the experimental rabbits (Figure 1). Hemorrhages were observed in various organs. The liver exhibits characteristic histopathological features of RHD, including apoptosis and variable lytic and coagulative hepatocellular necrosis. The hepatic lobule was indistinctly delimited, and the arrangement of liver cords appeared disordered (Figure 1A,B). Significant damage was observed in numerous hepatocytes, accompanied by cell atrophy and cytoplasmic or nuclear dissolution. Pulmonary lesions were observed, including bronchial epithelial cell exfoliation, a substantial eosinophilic serous exudation in the alveolar cavity, and a part of the alveolar cavity fusing to form a pulmonary bulla. Slight thickening of the alveolar diaphragm, vasodilation, and congestion were accompanied by lymphocyte and macrophage infiltration (Figure 1C,D). The red pulp of the spleen exhibited congestion, cellular lysis, and necrosis, accompanied by a significant presence of vacuoles (Figure 1E,F). The kidney showed glomerular congestion with substantial accumulation of red blood cells in the capillaries, mesangial cell lysis, and a fuzzy structure (Figure 1G,H). The histopathological examination revealed a loose arrangement of myofibers in the heart, with evidence of fragmentation and dissolution (Figure 1I,J).

### 3.2. Viral RNA Burden and Transcriptomic Pattern of Spleen

The trophic liver and spleen tissues were collected after infection with CHN/SC2020 at 16 hpi. The numbers of viral RNA copies in the liver, spleen, lung, kidney, and heart had an average value of 7.52 × 10^6^, 5.02 × 10^6^, 6.14 × 10^5^, 2.91 × 10^5^, and 1.52 × 10^5^ per 0.1 g of tissue, respectively (Appendix A). Since the viral burdens of the liver and spleen were comparative and the transcriptomic spectrum of the spleen was unclear, we subsequently focused on the molecular expression of immune-associated genes in the spleen. Therefore, the RNA-Seq technique was utilized to establish the sequencing libraries of gene expression in triplicate using the Illumina NovaSeq 6000 platform. After removing the low-quality reads, we obtained an average of 47,910,234 and 48,029,098 clean reads from the infected and non-infected group libraries, respectively. The proportion of clean reads in all the samples was greater than 97%, and the mapping rate of the clean reads in each sample was higher than 85%, thus demonstrating the reliability of the sequencing data quality (Table 2). Alignment analyses of the sequences from the infected and control samples were 86.96% and 86.93%, respectively, mapped to the Oryctolagus cuniculus genome (GenBank: GCF_009806435.1).

Gene expression levels were measured by short-read mapping and were expressed as reads per kilobase per million mapped (RPKM) adjusted by a normalization factor. We detected 20,809 expressed genes or transcripts in all six samples, and within each sample, 17,392 to 17,993 expressed genes or transcripts were detected, respectively (Table 2 and Appendix A). According to the transcriptome features, the standard uniquely mapped reads approach was adjusted by constructing the sequence clusters before reading mapping. The PCA results revealed that the transcriptome profiles of the samples subjected to the same treatment were clustered together (Appendix A), confirming the reproducibility of the transcriptomic sequencing in the different groups.

### 3.3. Differentially Expressed Gene (DEG) Screening in the Spleen

To explore the gene expression pattern of the spleen under RHDV2 infection, we compared the number of DEGs across treatment groups. A gene was deemed to be differentially expressed if the fold-change of the RPKM expression values was at least two and the divergence probability was at least 0.7. Using both comparisons for a library pair, 3148 genes were differentially expressed, with 1501 genes upregulated and 1647 genes downregulated (Figure 2, Appendix A).

### 3.4. Functional Analysis and Biological Enrichment of DEGs

A gene ontology (GO) enrichment analysis on each of the genes was performed to gain insight into the biological roles of the DEGs. The GO analysis revealed that the DEGs were enriched in many GO categories, including biological processes, cellular components, and molecular function (Appendix A). After the CHN/SC2020 infection, there was an increase in the number of DEGs in the spleen, which was involved in several biological processes. Among the upregulated genes, 662, 426, and 323 were mapped to ‘cellular component’, ‘molecular function’, and ‘biological process’, respectively (*p* < 0.05) (Figure 3). The main GO terms for DEGs included ‘cytosol’ (GO:0005829), ‘cytoplasm’ (GO:0005737), and ‘nucleoplasm’ (GO:0005654) of cellular components, ‘ATP binding’ (GO:0005524), ‘identical protein binding’ (GO:0042802), and ‘zinc ion binding’ (GO:0008270) of molecular functions, and ‘immune response’ (GO:0006955), ‘innate immune response’ (GO:0045087), and ‘inflammatory response’ (GO:0050729) of biological processes. We found eight GO terms related to inflammation, including ‘inflammatory response’ (GO: 0006954), ‘positive regulation of inflammatory response’ (GO: 0050729) and ‘leukocyte migration involved in inflammatory response’ (GO: 0002523) (Figure 3). We found that 6, 12, and 32 of the downregulated genes were associated with ‘integral component of membrane’ (GO:0016021), ‘cytoplasm’ (GO:0005737) and ‘extracellular region’ (GO:0005576) of cellular components, ‘calcium ion binding’ (GO:0005509), ‘identical protein binding’ (GO:0042802) and ‘transcription factor activity, sequence-specific DNA binding’ (GO:0003700) of molecular functions, and ‘signal transduction’ (GO:0007165), ‘intracellular signal transduction’ (GO:0035556) and ‘cell adhesion’ (GO:0007155) of biological processes.

KEGG analysis of the DEGs assigned after infection revealed that 116 of the upregulated genes were significantly enriched in 20 pathways, which were related to disease, signal transduction, and cytokine signaling (*p* < 0.05) (Figure 4, Appendix A). Nine DEGs showed cytokine–cytokine receptor interaction signaling pathway enrichment (KO: ocu04061); nine DEGs showed NOD-like report signaling pathway enrichment (KO: ocu04621); and nine DEGs showed Jak/STAT signaling pathway enrichment (KO: ocu04630). These pathways could activate signaling cascades to produce inflammation and mediate viral replication. These DEGs were involved in the pathways associated with disease, signal transduction, cytokine signaling, and cellular processes, all of which might be involved in viral pathogenesis. The DEGs involved in cytokine signaling were upregulated following infection with RHDV2 and included inflammatory cytokines such as IL1α, IL-6, and IL-8, and chemokines such as CCL2, CXCL9, and CXCL11, which were significantly upregulated compared with the non-infected rabbits.

Interestingly, the 947 downregulated genes were significantly enriched in 20 signaling pathways (*p* < 0.05). The DEGs were significantly enriched in the pathways in cancer (KO: ocu05200), the calcium signaling pathway (KO: ocu04020), and the Rap1 signaling pathway (KO: ocu04015) (Figure 4; Appendix A).

### 3.5. Transcriptome Data Verification by qRT-PCR

To further evaluate our DEG library, six upregulated DEGs (IL1α, IL-6, IL-8, IL-22, CCL2, and CXCL9) and three downregulated DEGs (NMUR1, HSPB7, and KCNIP2) were selected for qRT-PCR analysis. GAPDH was selected as the internal reference for the qRT-PCR experiments. The relative expression of the nine genes in the infected group compared to the control group is displayed in Table 3. Analysis revealed that the results of qRT-PCR showed the same patterns of expression as those observed with the DEGs data, confirming the reliability and accuracy of the sequencing results.

## 4. Discussion

In April 2020, the first RHDV2 outbreak occurred in rabbit farms located in the southwest of China [13,16,30]. Before that, the prevention and control of RHD had been effective and stable in local areas. However, the outbreak of RHDV2 caused substantial losses to the rabbit industry. As previously documented, the pathogenicity of the CHN/SC2020 strain to domestic rabbits was similar to that of other RHDV2 strains identified in mainland China [31], but biological differences were still observed amongst these isolates. In our study, all adult rabbits challenged by CHN/SC2020 died within 16–23 hpi, and most of them experienced acute death. The death duration seemed shorter than that of other RHDV2 strains [31,32], which might suggest diversified viral phenotypes within the same genotype. The hemorrhagic characteristics in the infected organs were consistent with the acute RHD process as observed in the liver, spleen, lung, and kidney tissues with severe pathological damage [17,25]. The increasing viral loads of the spleen were parallel to those of the trophic liver, indicating that the spleen was another arsenal for RHDV2 replication and an important mediator of immune dysregulation. These findings prompted us to uncover the potential activities of the spleen that modulated the infection’s immunity. It was well known that the spleen, the largest secondary lymphoid organ, consists of antigen-presenting cells (such as macrophages and lymphocytes) to capture invading pathogens, which could exert the innate immunity of the host against infection [19,20]. Accordingly, positive RHDV2 antigen immunolabelling had been widely identified in splenic macrophages and lymphocytes of the red pulp and in perifollicular areas, accompanied by an increasing number of macrophages and reduced lymphocytes [17].

In the current study, the RNA-Seq analysis of RHDV2-infected spleens confirmed the molecular mechanisms involved in viral infection, immune response, and cell signaling activation. The dysregulated DEGs (1501 upregulated and 1647 downregulated) suggested an active immune response of the spleen to RHDV2 infection. A series of the upregulated DEGs of interest converged in proinflammatory cytokine and chemokine (e.g., IL1α, IL-6, IL-8, IL-22, CCL2, CCL4, CCL5, CXCL9, CXCL11, and CXCL10) pathways contributing to virus-associated pathogenecity in the spleen, which were also confirmed by quantitative mRNA detection. These cytokines and chemokines were often induced by oxidant stress and viral infection, resulting in cell damage, chronic inflammation, and excessive immune responses [33].

Chemokines are a superfamily of small-molecule cytokines that can elicit a chemotactic response and promote the secretion of cell signaling proteins with broad-spectrum effects on innate immunity [34]. Thus, in early RHDV2 infection, chemokines might recruit a variety of white blood cells to the inflammatory sites, which was confirmed in the pathohistological findings. Moreover, interleukins belong to a class of cytokines that have multiple effects and play critical immunomodulatory roles in recruiting immune cells to establish an inflammatory microenvironment. But high levels of cytokines and chemokines would trigger a “cytokines storm” that disrupted the balance between innate and adaptive immune responses [35]. Similar phenomena had been found in the impaired spleen by other RNA viruses [35,36].

In several studies, levels of IL-1A, IL-6, IL-8, IL-22, CCL-2, and CXCL-9 rapidly increased in innate immune cells, such as macrophages, a component of host defense to clear infected cells when encountering damage-associated molecular patterns (DAMP) or pathogen-associated molecular patterns (PAMPs) [33,34,37]. However, the overproduction of cytokines would conversely deteriorate systemic inflammation and immune malfunction [36]. Correspondingly, RHDV2 infection induced a high expression level of cytokines (such as IL-1, IL-6, IL-8, IL-18, and TNF-α), which contributed to the inflammatory cascade and progression of RHD [38,39]. A similar situation had also been identified in the fatal African swine fever virus (ASFV) infection, which boosted the upregulation of IL-1, IL-6, and TNF and chemokines (CCL2, CCL5, and CXCL10) and cellular apoptosis and pyroptosis [35]. The highly pathogenic RHDV2 is expected to disrupt the immune homeostasis of the spleen, thereby impairing its proper functioning [40].

Novelly, the neuromedin U receptor 1 (NMUR1), which is expressed on activating group 2 innate lymphoid cells, mediated the type 2 immune responses in host defense [41]. Its downregulated expression suggested that it inhibited the innate immune response against RHDV2 infection, and this finding might be partly attributed to the decreased lymphocytes in the spleen [17]. Similarly, the downregulation of voltage-gated potassium (Kv) channel-interacting protein 2 (KCNIP2) functioning as a Ca^2+^ signaling-dependent transcriptional repressor was also observed, which might give rise to chemokines and cytokines upregulation [36], and might promote RHDV2-associated spleen dysfunction. Additionally, HSPB7 (small heat shock protein family B member 7) is known to respond to stressors such as hypoxia and viral infection. It plays a crucial role in cellular innate immunity, viral protein translation, and transport [42]. Its downregulation might potentially compromise the host’s defense against early infection.

A recent study focused on the transcriptomic pattern of liver in rabbit kittens challenged by RHDV2 (BlMt-1 strain) showed that the virus could replicate rapidly by suppressing host innate immune responses at the early infection (12 hpi and 24 hpi) [18]. Our research revealed that DEGs associated with proinflammatory cytokines and chemokines were upregulated in late infection. These findings suggested that an abnormal immune response to the RHDV2 invasion was persistent during the ongoing virus infection. At the same time, the excessive activation of inflammatory signals in the spleen might result in perturbation of host immunity, thereby facilitating the rapid replication of RHDV2 and subsequent mortality. The limitation of the current study was that it was not comprehensive enough to observe the dynamic change of the splenic transcriptome and immune cells’ phenotype, which might be helpful in unveiling the molecular mechanism of RHDV2 pathogenicity. The trade-off game between the virus and host can be further elucidated through temporal and spatial analysis of the splenic transcriptome, thereby enhancing our understanding in a more specialized manner.

## 5. Conclusions

In this study, we investigated the pathogenicity of RHDV2-CHN/SC2020 and analyzed the transcriptomics of spleen tissue. CHN/SC2020 exhibited robust pathogenicity, eliciting a significant upregulation in the expression of numerous genes associated with disease, signal transduction, cellular processes, and cytokine signaling categories. Notably, there was an upregulation in the expression of cytokines and chemokines involved in inflammation. These findings suggested that viral infection could disrupt the cytokine network within the spleen and lead to inflammatory disorders. This study provided a novel insight into the splenic immune response induced by RHDV2 infection, thereby expanding our understanding of RHDV2 pathogenicity.

## Figures and Tables

**Figure 1 viruses-16-00495-f001:**
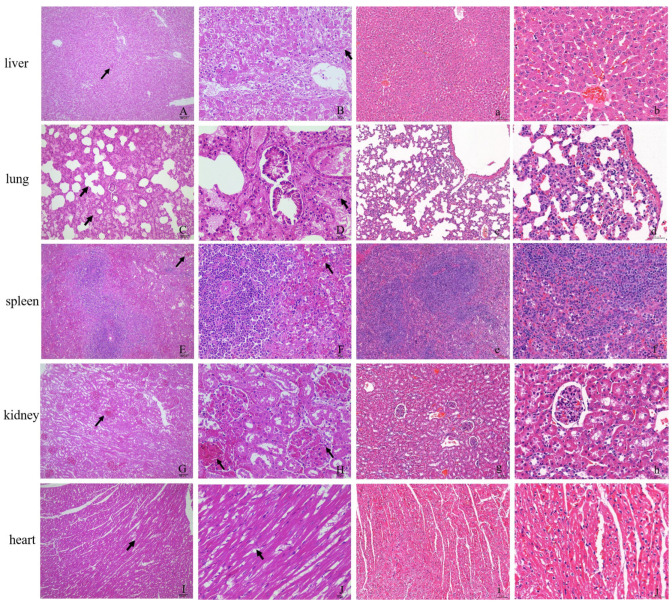
Histopathological changes in RHDV2/CHN/SC2020-infected rabbits. (**A**–**J**): Hematoxylin and eosin (H&E) section staining of the organs from RHDV2/SC2020-infected rabbits. (**a**–**j**): H&E section staining of the organs from the control rabbits. (**A**,**B**) The livers of infected rabbits exhibited disordered liver cords and necrotic hepatocytes (arrow). (**C**,**D**) Alveolar cavity fusion formed bullae and eosinophilic serous exudates in the cavity; alveolar epithelial cell lysis in the lung of infected rabbits (arrow). (**E**,**F**) The red pulpous splenic sinus congestion in the spleen (arrow). (**G**,**H**) Congestion and cytolysis in the kidney (arrow). (**I**,**J**) The myofibers were loosely arranged and with rupture in the heart. 100× magnification and 400× magnification.

**Figure 2 viruses-16-00495-f002:**
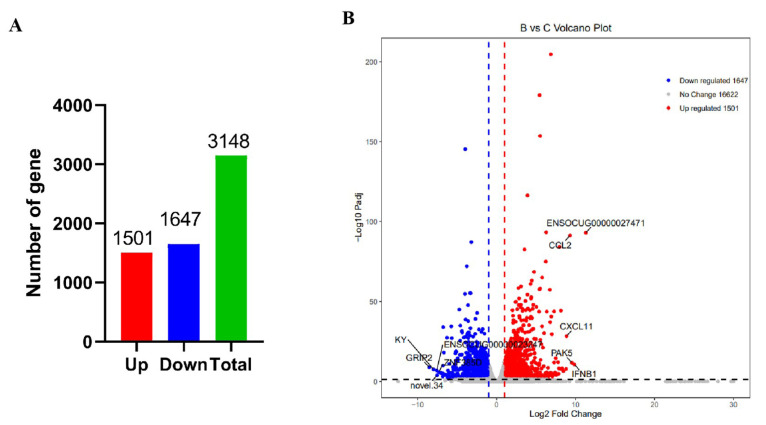
The genes that were differentially expressed after infection with RHDV2/CHN/SC2020. (**A**) The number of DEGs. (**B**) Volcano plot showing the DEGs.

**Figure 3 viruses-16-00495-f003:**
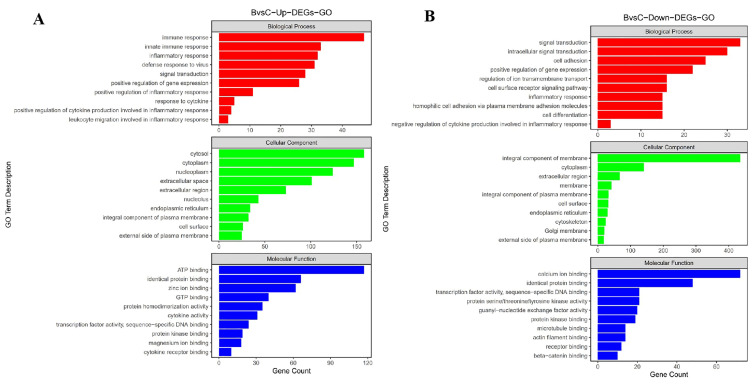
GO assignments for DEGs in the spleen after infection with RHDV2/CHN/SC2020. GO categories were shown for upregulated genes (**A**) and for downregulated genes (**B**). The GO categories included biological processes, cellular components, and molecular functions. The x-axis showed the GO categories, and the y-axis showed the number of genes.

**Figure 4 viruses-16-00495-f004:**
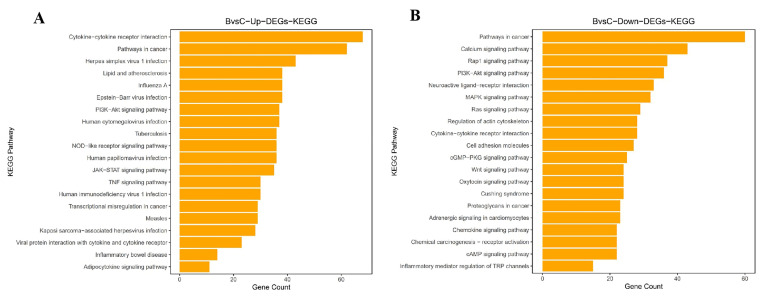
KEGG classification of the DEGs post RHDV2 infection. The KEGG classifications for the upregulated (**A**) and downregulated (**B**) genes were shown. The *x*-axis indicated the pathway, and the *y*-axis indicated the number of DEGs.

**Table 1 viruses-16-00495-t001:** Mortality rate of CHN/SC2020-infected rabbits.

Group	Time of Death Postinfection (PI)	Clinical Symptoms	Number of Deaths	Mortality (%)
Infection	between 12/16 h	anorexia, lethargy	1	100%
	between 17/20 h	anorexia, lethargy	1	
	between 21/24 h	anorexia, lethargy, bleeding from the nose	3	
Control	-	-	none	0%

**Table 2 viruses-16-00495-t002:** Summary of the sequencing reads from the spleens with and without CHN/SC2020 infection.

Sample Group	Sample Name	Clean Data (bp)	Clean Reads Number	Clean Rate (%)	Genome Mapping (%)	Detected Gene NO.
Infection group	B	6,791,407,800	45276052	96.48%	87.74%	17,552
		7,982,321,400	53215476	97.79%	86.72%	17,532
		6,785,876,400	45239176	97.14%	86.42%	17,392
Control group	C	6,863,618,100	45757454	96.97%	86.96%	17,963
		6,705,405,900	44702706	97.82%	86.31%	17,829
		8,044,070,100	53627134	97.39%	87.52%	17,993

**Table 3 viruses-16-00495-t003:** Validation of the expression patterns of nine differentially expressed genes by qRT-PCR analysis.

Differentially Expressed Genes	IL1α	IL-6	IL-8	IL-22	CCL2	CXCL9	NMUR1	HSPB7	KCNIP2
qRT-PCR (2^−ΔΔct^)	4.85 ± 1.08	27.15 ± 3.11	19.72 ± 16.44	6.24 ± 1.88	122.42 ± 28.51	7.75 ± 5.07	0.28 ± 0.07	0.19 ± 0.12	0.27 ± 0.07
RNA-seq (log2)	4.97	11.31	6.35	6.58	9.30	6.91	−5.81	−6.18	−6.08
expression pattern	up	up	up	up	up	up	down	down	down

## Data Availability

Raw RNA-seq reads have been deposited in the NCBI Sequence Read Archive under SRA database: accession PRJNA1066492.

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
