# Peer review of "Neglected Spleen Transcriptional Profile Reveals Inflammatory Disorder Conferred by Rabbit Hemorrhagic Disease Virus 2 Infection"

_viruses, 2024, doi:10.3390/v16040495_

Round 1

Reviewer 1 Report

Comments and Suggestions for Authors

In this manuscript, the authors analyzed the splenic transcriptome and differential expression of genes of rabbits infected with RHDV2 used RNA-Seq technology, the finding provides novel insights into the splenic immune response induced by RHDV2. But some minor revisions that should be addressed:

1. In the text, the description format needs to be standardized. For instance, please unify the name description of the virus, check line 29, line 111, line 400.

2. It is better to explain why selected the spleen at 16 hpi for transcriptome analysis.

3. It is better to increase font size or clarity in fig 2B, fig 3A/B, fig 4A/B.

4. In Tab 3, the qRT-PCR result of GAPDH should be included.

5. Line 95 .What is the rationale for choosing the challenge dose.

6. qRT-PCR data of other organs should be supplemented.

7. Line 116. “Twenty to 30 mg” ? Check it

8. Line 174, rabbit adults not rabbit kits

9. Line177-178, please indicate the origin of reagents and equipment.

10. Please check the full text for spelling mistakes, Line 371, cellular.

11. As a general comment, the use of the English language could be further improved for a better readability of the manuscript and better comprehension of the work presented.

12. The authors should check the reference style in the entire manuscript. Such as size of character, Journal abbreviations, etc. Check Line 430,432, 452-453,474.

Comments on the Quality of English Language

As a general comment, the use of the English language could be further improved for a better readability of the manuscript and better comprehension of the work presented.

Author Response

In this manuscript, the authors analyzed the splenic transcriptome and differential expression of genes of rabbits infected with RHDV2 used RNA-Seq technology, the finding provides novel insights into the splenic immune response induced by RHDV2. But some minor revisions that should be addressed:

Comment 1. In the text, the description format needs to be standardized. For instance, please unify the name description of the virus, check line 29, line 111, line 400.

Response: We thank to the reviewer’s suggestions. The nomenclature of virus has been standardized throughout the manuscript.

Comment 2. It is better to explain why selected the spleen at 16 hpi for transcriptome analysis.

Response: According to the pathogenicity results, the death of infected rabbits occurred at approximately 16 hpi as a indicative of late stage of RHDV2 infection, when we accomplish spleen sampling for transcriptome analysis, .

Comment 3. In Tab 3, the qRT-PCR result of GAPDH should be included.

Response: The relative expression levels of target genes are calculated using the 2−ΔΔCt method normalized based on the GAPDH, thus the raw Ct values of GAPDH are not shown.

Comment 4. Line 95 .What is the rationale for choosing the challenge dose.

Response: The challenge dose is determined by referring to the literatures and viral load in the liver from infected rabbits.

Comment 5. qRT-PCR data of other organs should be supplemented.

Response: We are grateful to the reviewer’s advice. We have supplemented the data of viral loads bearing in other organs. The sentence “The numbers of viral RNA copies in liver, spleen, lung, kidney, and heart were an average value of 7.52 × 106, 5.02× 106, 6.14 × 105, 2.91× 105, and 1,52 × 105 per 0.1 g of tissue, respectively” was added to lines 230-232 in the revised manuscript.

Comment 6. Line 116. “Twenty to 30 mg” ? Check it

Response: We have corrected this error, and carefully checked the typo's and descriptions across the manuscript. The sentence “About 100 mg of each sample was homogenized with a tissue master.” was added to lines 119-120 in the revised manuscript.

Comment 7. Line 174, rabbit adults not rabbit kits

Response: We have fixed the wrong description.

Comment 8. Line177-178, please indicate the origin of reagents and equipment.

Response: We have described the information of producers for the reagents and equipment employed in this study. The sentence “For the qRT-PCR analysis, the TB Green® Premix Ex Taq™ II (Tli RNaseH Plus) Kit (TaKaRa, China) and the QuantStudio 3 real-time PCR system (ThermoFisher, USA) were used according to each manufacturer’s instructions.” was added to lines 181-184 in the revised manuscript.

Comment 9. Please check the full text for spelling mistakes, Line 371, cellular.

Response: We appreciate the reviewer's advice, we have revised the grammar or spelling errors throughout the text.

Comment 10. As a general comment, the use of the English language could be further improved for a better readability of the manuscript and better comprehension of the work presented.

Response: We appreciate this advice. The revised manuscript has been submitted for polishing by language service to make it more readable to general audience.

Comment 11. The authors should check the reference style in the entire manuscript. Such as size of character, Journal abbreviations, etc. Check Line 430,432, 452-453,474.

Response: The reference style and format have been revised according to the publishing guidelines.

Reviewer 2 Report

Comments and Suggestions for Authors

Yu et al., determine in this manuscript the splenic transcriptome and differential gene expression in naïve and RHDV2-infected rabbits using RNA-Seq. This is the first transcriptional investigation of the spleen after RHDV2 challenge, which represents a certain degree of novelty. RHD is a viral disease that primarily affects the liver, but the viral antigen can also be detected in splenic macrophages. In fact, the viral strain used in this study, RHDV2-CHN/SC2020, frequently damages the spleen in addition to the liver, making this study relevant. This manuscript is clearly written and well organized. The figures and tables are appropriate and informative.  

The results provide a reasonable number of potential molecules/pathways contributing to RHDV2 pathogenesis, to pursue and refine in future studies.  The obvious limitation of the study, that the authors acknowledge, concern the dynamic changes of splenic transcriptome and the analyses of cell phenotype, required to confirm the mechanisms of RHDV2 pathogenesis or immune response evasion.

This reviewer recommends accepting the manuscript for publication after extensive proofreading of typos.

Author Response

Yu et al., determine in this manuscript the splenic transcriptome and differential gene expression in naïve and RHDV2-infected rabbits using RNA-Seq. This is the first transcriptional investigation of the spleen after RHDV2 challenge, which represents a certain degree of novelty. RHD is a viral disease that primarily affects the liver, but the viral antigen can also be detected in splenic macrophages. In fact, the viral strain used in this study, RHDV2-CHN/SC2020, frequently damages the spleen in addition to the liver, making this study relevant. This manuscript is clearly written and well organized. The figures and tables are appropriate and informative.

The results provide a reasonable number of potential molecules/pathways contributing to RHDV2 pathogenesis, to pursue and refine in future studies. The obvious limitation of the study, that the authors acknowledge, concern the dynamic changes of splenic transcriptome and the analyses of cell phenotype, required to confirm the mechanisms of RHDV2 pathogenesis or immune response evasion.

This reviewer recommends accepting the manuscript for publication after extensive proofreading of typos.

Response: We appreciate the approval by the reviewer. We will further strengthen the scientific rigor based on the findings in current study.

Reviewer 3 Report

Comments and Suggestions for Authors

The paper sounds very interesting by the subject promising providing novel insights into the splenic immune response induced by RHDV GI.2 infection, but the insight of the paper is dissappointing. There are several serious flaws of the paper:

- very small number of animals included on the study;

- rapid death of the animals, which makes is difficult to draw conclusions on the immune responses;

- mistakes in the text;

- very modest manner of describing the methods;

- several mistakes in citations;

- laconic conclusion.

Therefore I do not recommend the paper to be published.

Comments on the Quality of English Language

The paper sounds very interesting by the subject promising providing novel insights into the splenic immune response induced by RHDV GI.2 infection, but the insight of the paper is dissappointing. There are several serious flaws of the paper:

- very small number of animals included on the study;

- rapid death of the animals, which makes is difficult to draw conclusions on the immune responses;

- mistakes in the text;

- very modest manner of describing the methods;

- several mistakes in citations;

- laconic conclusion.

Therefore I do not recommend the paper to be published.

Author Response

The paper sounds very interesting by the subject promising providing novel insights into the splenic immune response induced by RHDV GI.2 infection, but the insight of the paper is dissappointing. There are several serious flaws of the paper, therefore I do not recommend the paper to be published.

Comment 1. very small number of animals included on the study.

Response:We appreciate the reviewer’s comment. In this study, a total of 16 rabbits were enrolled. Among them, 6 infected rabbits were submitted for further RNA-sequence analysis, which is at least considered to be enough for statistical analysis, although more numbers of animals involved may further substantiate the findings.

Comment 2. rapid death of the animals, which makes is difficult to draw conclusions on the immune responses;

Response: It has been revealed in other studies that RHDV2 infection can trigger robust immune responses in early infection, even though rapid death occurs in infected rabbits, which may be helpful for us to investigate the alteration of transcriptional factors focusing splenic immune dysregulation.

Comment 3. mistakes in the text;

Response: We have carefully checked and corrected the spelling or grammar mistakes in the text.

Comment 4. very modest manner of describing the methods;

Response: We have rephrased the methodologies in this study, which may be helpful for the interpretation of findings in this study.

Comment 5. several mistakes in citations;

Response: We have corrected the mistakes in citations.

Comment 6. laconic conclusion. 

Response: The authors have rewritten the conclusion and made the statements more reasonable. Altogether, we will endeavor to improve the manuscript as suggested to achieve a appropriate version for approval.

The sentence "In this study, we investigated the pathogenicity of RHDV2-CHN/SC2020 and analyzed the transcriptomics of spleen tissue. CHN/SC2020 exhibited robust pathogenicity, eliciting a significant upregulation in the expression of numerous genes associated with disease, signal transduction, cellular process and cytokine signaling categories. Notably, there was an upregulation in the expression of cytokines and chemokines involved in inflammation. These findings suggested that viral infection could disrupt the cytokine network within the spleen, and led to inflammatory disorder." was added to lines 406-412 in the revised manuscript.

Round 2

Reviewer 3 Report

Comments and Suggestions for Authors

Thank you for the corrections, nevertheless in my opinion the paper still is not reaching the standards. Primary I recommended rejection and I do not change my mind, sorry.